# Functional Characteristics of Diverse *PAX6* Mutations Associated with Isolated Foveal Hypoplasia

**DOI:** 10.3390/genes14071483

**Published:** 2023-07-21

**Authors:** Itsuka Matsushita, Hiroto Izumi, Shinji Ueno, Takaaki Hayashi, Kaoru Fujinami, Kazushige Tsunoda, Takeshi Iwata, Yoshiaki Kiuchi, Hiroyuki Kondo

**Affiliations:** 1Department of Ophthalmology, University of Occupational and Environmental Health, Kitakyushu 807-8555, Japan; i-matsushita@med.uoeh-u.ac.jp; 2Department of Occupational Pneumology, University of Occupational and Environmental Health, Kitakyushu 807-8555, Japan; h-izumi@med.uoeh-u.ac.jp; 3Department of Ophthalmology, Hirosaki University Graduate School of Medicine, Hirosaki 036-8562, Japan; uenos@hirosaki-u.ac.jp; 4Department of Ophthalmology, Nagoya University Graduate School of Medicine, Nagoya 466-8550, Japan; 5Department of Ophthalmology, The Jikei University School of Medicine, Tokyo 105-8461, Japan; taka@jikei.ac.jp; 6Department of Ophthalmology, National Hospital Organization Tokyo Medical Center, Tokyo 152-8902, Japan; k.fujinami@ucl.ac.uk (K.F.); kazushige.tsunoda@kankakuki.jp (K.T.); 7Division of Molecular and Cellular Biology, National Institute of Sensory Organs, National Hospital Organization Tokyo Medical Center, Tokyo 152-8902, Japan; takeshi.iwata@kankakuki.jp; 8Department of Ophthalmology and Visual Science, Graduate School of Biomedical and Health Sciences, Hiroshima 734-8551, Japan; ykiuchi@hiroshima-u.ac.jp

**Keywords:** foveal hypoplasia, FVH1, aniridia, corneal opacity, *PAX6*, DNA binding, transcriptional activity, paired domain, proline-serine-threonine-rich domain

## Abstract

The human fovea is a specialized pit structure in the central retina. Foveal hypoplasia is a condition where the foveal pit does not fully develop, and it is associated with poor vision. Autosomal dominant isolated foveal hypoplasia (FVH1) is a rare condition of foveal hypoplasia (FH) that lacks any other ocular manifestations. FVH1 is associated with hypomorphic mutations in the *PAX6* gene that encodes a sequence-specific DNA-binding transcription factor for morphogenesis and evolution of the eye. We report our findings in 17 patients with *PAX6* mutations associated with FVH1 or FH with aniridia and corneal opacities. Patients with three mutations, p.V78E, p.V83F and p.R128H, in the C-terminal subdomain of the paired domain (CTS) consistently have severe FH. Luciferase assays for a single reporter containing a representative PAX6 binding site indicated that the transcriptional activities of these mutations were significantly reduced, comparable to that of the truncation mutation of p.G65Rfs*5. Patients with p.P20S in the N-terminal subdomain of the paired domain, and a patient with p.N365K in the proline-serine-threonine-rich domain (PSTD) had mild FH. A patient with p.Q255L in the homeodomain had severe FH. The P20S and Q255L mutants did not affect the transcriptional activity. Mutant N365K has a retained DNA-binding activity but a reduced transcriptional activity, due to a low PSTD transactivation. These findings demonstrated that mutations associated with FVH1 underlie a functional divergence between DNA-binding ability and transcriptional activity. We conclude that a wide range of mutations in the *PAX6* gene is not limited to the CST region and are responsible for FVH1.

## 1. Introduction

The fovea is a specialized pit structure in the central retina. The presence of a high density of cones at the bottom of the foveal pit makes it the point with high visual acuity [1]. Foveal hypoplasia is a retinal disorder in which there is a lack of a complete development of the morphology of the fovea, and it is associated with poor vision. Several disorders are known to be associated with foveal hypoplasia, including aniridia (OMIM 106210) and albinism (OMIM 203200) [2]. It is known that mutations of the *PAX6* gene can cause aniridia. The product of the *PAX6* gene is a transcription factor that is essential for the development of the eyes of vertebrates and non-vertebrates, and it is considered to be a master regulator for the development of the eye [3]. This transcription factor is essential for the development of the normal structure of the retinal tissues, as well as other eye tissues including the lens and corneal epithelium [4]. The *PAX6* gene is also expressed in the central nervous system [5].

There are numerous isoforms of the *PAX6* genes, of which two major isoforms, canonical *PAX6* and *PAX6*(5a), play a pivotal role in the genesis of the eye. The canonical PAX6 protein encodes a protein of approximately 46 kDa consisting of 422 amino acids. The PAX6(5a) has an additional 14 amino acids inserted between exons 5 and 6. The canonical PAX6 and PAX6(5a) isoforms have different DNA-binding properties, and serve as a molecular switch that specifies the target genes [6]. The PAX6(5a) isoform is highly expressed in the fovea, and it is associated with the formation of the fovea by interacting with the canonical PAX6 isoform [7]. These isoforms have two DNA-binding domains, a paired domain (PD) and a homeodomain (HD), followed by the proline-serine-threonine-rich domain (PSTD) [8,9]. PD is divided into the N-terminal subdomain (NTS) and the C-terminal subdomain (CTS). These subdomains bind to the DNA sequences of the target genes to regulate their transcriptional activity [10,11]. 

Isolated foveal hypoplasia is a rare condition that lacks any other ocular manifestations. The autosomal dominant form of the isolated foveal hypoplasia (FVH1, OMIM 136520) is known to be associated with hypomorphic mutations of the *PAX6* gene. Patients with FVH1 present with poor visual acuity and nystagmus. At present, approximately 680 different mutations in the *PAX6* gene are known to cause aniridia, while only 12 different mutations of the *PAX6* gene are associated with isolated foveal hypoplasia (Human Gene Mutation Database, Professional version 2023.1, https://portal.biobase-international.com/hgmd/pro/star/php, accessed on 1 June 2023). The *PAX6* mutations associated with isolated foveal hypoplasia have been found to be in the CTS region, which suggests the critical role of the CTS plays in foveal development [12,13]. However, a few mutations were reported to be located in loci other than the CTS, and their functional relevance remains unknown [14]. 

We have recently reported our findings on two patients with *PAX6* mutations associated with FVH1 [15]. Both mutations were located in loci other than the CTS, i.e., one was in the NTS and the other in the PSTD. The purpose of this study was to determine the characteristics of nine Japanese families with FVH1 and the other *PAX6*-related phenotypes with foveal hypoplasia. The genotype–phenotype correlation of the *PAX6* mutations was investigated using functional studies.

## 2. Materials and Methods

This was a multicenter retrospective case series study. The procedures used conformed to the tenets of the Declaration of Helsinki, and they were approved by the following: the Ethics Committee of the University of Occupational and Environmental Health Japan (Project identification code 20-148, approved on 18 January 2021); Nagoya University (Project identification code 2010-1067, approved on 1 April 2010); the Jikei University School of Medicine (Project identification code 24-231 6997, approved on 3 December 2012); and the National Hospital Organization Tokyo Medical Center (Project identification code R22-046, approved on 28 July 2022). A signed informed consent was obtained from all the patients or their parents

### 2.1. Clinical Examination

We studied seventeen patients from nine families (mean age 24 ± 21 years; range 1 to 68 years) and six families were diagnosed with FVH1. The findings of two of these were reported earlier [15], two families with aniridia with corneal opacities, and one family with a peripheral corneal opacity and mild iris anomaly (Table 1). The ocular examinations included measurements of the refractive error and visual acuity, and examinations using slit-lamp biomicroscopy, gonioscopy, and ophthalmoscopy. Optical coherence tomographic (OCT) images were recorded from all patients. OCT angiography (OCTA) images were obtained from selected patients with good visual acuity, and Patient 7 who underwent examinations under general anesthesia using SS-OCT (DRI OCT Triton, Topcon, Tokyo, Japan), and Patient 13 using spectral domain OCT (RTVue XR Avanti, Optovue, Inc., Fremont, CA, USA).

### 2.2. DNA Sequence Analysis

Genomic DNA was extracted from the peripheral blood using DNA extraction kits. Polymerase chain reaction (PCR) followed by Sanger sequencing and/or whole exome sequencing was performed on the coding exons of the *PAX6* gene. A detailed description of the sequencing procedures has been presented [15,16]. Reference sequences of *PAX6* (NM_000280.4) were used with variations numbered based on its cDNA sequence, with +1 corresponding to the first nucleotide of the initiation codon (ATG). The allelic frequency of the variants was searched for in the databases of the Japanese population (Human Genetic Variation Database, HGVD, and the Tohoku Medical Megabank Organization database, Tommo3, or other population databases of the 1000 genomes project and the Genome Aggregation Database, gnomAD [17,18,19,20]. A conservation of the amino acid residues among humans and other species, e.g., rhesus monkey, mice, elephant, chicken, zebrafish, and frog) was assessed by the UCSC Genome Browser [21]. The pathogenicity of the variants was predicted by six in silico programs [22,23,24,25,26,27]. Finally, the pathogenicity of the variants was determined based on the recommendation of the American College of Medical Genetics and Genomics (ACMG) standard and guidelines [28].

### 2.3. Functional Assay

#### 2.3.1. Cell Lines and Culture

The human retinal pigment epithelium cell line, ARPE19/HPV16, the human prostate cancer cell line, PC3, and the monkey kidney fibroblast-like cell line, COS1, were purchased from the American Type Culture Collection (Manassas, VA, USA). The immortalized human dopaminergic neuronal precursor cell line, LUHMES, was obtained from Applied Biological Materials Inc. (Richmond, BC, Canada). The PC3 cells and the COS1 cells were cultured in DMEM. The LUHMES cells and the ARPE19/HPV16 cells were cultured in DMEM/Ham’s F-12 medium with N2 supplement (Cat No. 17502048, ThermoFisher Scientific, Waltham, MA, USA) and DMEM/F12 (1:1 mixture of Dulbecco’s modified Eagle’s medium and Ham’s F12), respectively. All media were supplemented with 10% heat-inactivated fetal bovine serum (FBS) and 1% (*v*/*w*) penicillin/streptomycin, and cells were maintained at 37 °C with 5% CO_2_.

#### 2.3.2. Antibodies

The anti-DYKDDDDK antibody (019-22394) conjugated with HRP was obtained from Fujifilm (Tokyo, Japan). The anti-β-actin antibody (A3854) conjugated with HRP was obtained from Sigma-Aldrich (Burlington, MA, USA), and the anti-GAPDH antibody (sc-47724) conjugated with HRP and anti-Lamin A/C antibody (sc-7293) were obtained from Santa Cruz (Santa Cruz, CA, USA).

#### 2.3.3. Plasmid Preparation

The expression and reporter plasmids of pTA-5aCON-Luc and pCE-Flag-PAX6(5a) were kind gifts from Dr. Nishina, Department of Development and Regenerative Biology, Medical Research Institute, Tokyo Medical and Dental University, Tokyo [29]. A DNA fragment containing a single copy of 5aCON [12], with a minimal promoter excised from pTA-5aCON-Luc and a poly A signal sequence was inserted upstream and downstream of nanoluciferase (Nluc), obtained from pNL1.1 (Promega, Madison, WI, USA). The resulting DNA fragment was then placed in the lentiviral expression plasmid pCDH GFP-T2A-Puromycin from Sigma-Aldrich. This resulted in the pCDH 5aCON-Nluc G2P plasmid and, similarly, a DNA fragment containing nine binding sites for GAL4 and adenovirus.

A major late promoter sequence excised from pGL4.35 luc2P 9xGAL4UAS_Hygro (Promega) and a poly A signal sequence were s inserted upstream and downstream of nanoluciferase (Nluc), respectively. The resulting DNA fragment was inserted into the lentiviral expression plasmid pCDH RFP-T2A-Puromycin (System Biosciences, Palo Alto, CA, USA), resulting in the pCDH 9xGAL4UAS-Nluc R2P plasmid. PAX6(5a) cDNA was excised from pCE-Flag-PAX6(5a) and inserted into the pcDNA3 RFP plasmid. Each missense mutant identified in this study, viz.,P20S, V78E, V83F, R128H, Q255L, V256E, N365K and the other mutants found in a patient with aniridia (G65Rfs*5), were generated using the PrimeSTAR Mutagenesis Basal Kit (Takara Bio, Kusatu, Japan). To obtain pcDNA3 GAL4-DYK-PAX6-TA(WT) RFP and pcDNA3 GAL4-DYK-PAX6-TA(N379K) RFP, the PSTD region (269–422 aa) excluding the DNA-binding region was amplified by PCR and added downstream of GAL4-Flag [30]. The resulting DNA fragment was then placed in the pcDNA3 RFP plasmid. Oligonucleotides were used to generate the mutants, and the deletion constructs are shown in Table 2.

#### 2.3.4. Establishment of Lentivirus and Stable Expression Cells

Non-proliferative lentivirus was packaged using the lentivirus packaging system purchased from the System Biosciences. Briefly, the pCDH 5aCON-Nluc G2P or pCDH 9xGAL4UAS-Nluc R2P plasmids were mixed with a pPACKH1 plasmid mix consisting of pPACKH1-GAG, pPACKh1-REV, and pVSV-G (System Biosciences), and then transfected into COS1 cells using X-tremeGENE 9 DNA Transfection Reagent (Roche, Basel, Switzerland). After 36 h, the culture medium was filtered through a 0.45 µm syringe filter and directly added to the PC3 cells not expressing PAX6. The infected cells were then cultured with 10 µg/mL puromycin (ant-pr; InvivoGen, San Diego, CA, USA) for more than 2 weeks to establish the PC3-5aNluc and PC3-9Gal4Nluc cells.

#### 2.3.5. RNA Preparation and Quantitative RT-PCR

Total RNA was purified using the miRNeasy Kit (217084; Qiagen, Germantown, MD, USA). qRT-PCR was performed as described [31] using the following primer sets: Hs01088114_m1 for *PAX6*, and Hs01060665_g1 for β-actin (Applied Biosystems, Foster City, CA, USA). The values were normalized to human β-actin expression. The comparative cycle time method was used to quantify the gene expression, and all samples were analyzed in duplicate.

#### 2.3.6. Transfection of Expression Plasmid

A total of 1 × 10^5^ PC3-5aNluc, PC3-9Gal4Nluc, and PC3 cells were seeded into a 35 mm dish. The following day, 5 µg of plasmid expressing each PAX6 protein was transfected into the cells using X-treme GENE 9. After 48 h, the cells were washed twice with PBS and collected. Samples for the reporter assays and pull-down assays were immediately analyzed, while the remaining samples were stored at −80 °C for Western blot analysis.

#### 2.3.7. Reporter Assay

After collecting the cells, 200 µL of Reporter Lysis Buffer (Promega) was added to the cell pellet and sonicated for 10 s. The cell lysate was then centrifuged at 10,000× *g* for 2 min, and 100 µL of the resulting supernatant was mixed with 100 µL of the reagent from the Nano-Glo Luciferase Assay System (Promega) in a 96-well white plate. The light intensity was measured using a luminometer (Luminescencer JNII RAB-2300; Atto, Tokyo, Japan).

#### 2.3.8. Cell Fractionation and Pull-Down Assay

The cell pellets for the reporter assay were resuspended in 200 µL of hypotonic buffer A containing 10 mM Hepes–KOH (pH 7.9), 10 mM KCl, 0.1 mM EDTA–NaOH (pH 8.0), 0.1 mM EGTA, 1 mM dithiothreitol (DTT), and 0.5 mM phenylmethylsulfonyl fluoride (PMSF). The suspension was incubated on ice for 15 min. Nonidet-P40 was added to a final concentration of 0.25% (5 µL of 10% stock solution), and the cells were gently resuspended and then centrifuged at 4200× *g* for 5 min. The resulting supernatant was collected as the cytoplasmic fraction. For the nuclear pellet, two different methods were used for the analysis. To determine the nucleoplasmic to cytoplasmic ratio of PAX6 by Western blot, 205 µL of hypotonic buffer was added to the nuclear pellet to equalize the volumes of the nucleoplasmic and cytoplasmic fractions. Both fractionated samples were sonicated for 15 s and then centrifuged at 21,000× *g* for 10 min. Then, 40 µL of each supernatant, the nucleoplasmic (Nuc) and cytoplasmic (Cyt) fractions, were used for Western blot analysis.

For the pull-down assays, three dishes were prepared for one condition. The nuclear pellet was resuspended in 150 µL of high salt buffer C containing 20 mM Hepes–KOH (pH 7.9), 0.4 M NaCl, 1 mM EDTA–NaOH, 1 mM EGTA, 1 mM DTT, and 0.5 mM PMSF, and incubated for 30 min on ice. The suspension was then centrifuged at 21,000× *g* for 10 min, and the resulting supernatant was used for the pull-down assay (50 µL) as well as input (20 µL). Single-stranded oligonucleotides (see Table 2) were annealed, and biotinylated oligonucleotides were bound to M280 magnetic beads (Veritas) in buffer K, which contained 100 mM KCl, 10 mM Tris-HCl (pH 7.4), 0.05% NP-40, and 10% glycerol. The bound oligonucleotides were washed three times with buffer K using a magnetic stand and mixed with 50 µL of each Nuc fraction and 800 µL of buffer K. After binding for 30 min at room temperature using a rotating disk device, the samples were washed five times with buffer K, using a magnetic stand. The resulting pellet was suspended in 40 µL of RIPA buffer and used for subsequent Western blotting.

#### 2.3.9. Western Blot and Slot Blot

Next, 10 µL of 2× loading buffer and 3 µL of 1 M DTT were added to the sample prepared for the reporter assay, cell fractionation, and pull-down analysis. The mixture was boiled for 5 min and then subjected to SDS-polyacrylamide gel electrophoresis (SDS-PAGE). The separated proteins were transferred onto polyvinylidene difluoride (PVDF) membranes, and the respective primary antibodies, anti-Flag-HRP, anti-β-HRP, anti-GAPDH-HRP, and anti-Lamin A/C, were diluted as follows: anti-Flag and anti-β-actin antibodies were diluted 1/10,000, anti-GAPDH antibody was incubated at 1/1000 dilution for 1 h, and anti-Lamin A/C antibody was incubated at 1/1000 dilution for 1 h. Secondary anti-mouse IgG antibody conjugated HRP against anti-Lamin A/C antibody was diluted at 1/7500 and incubated for 45 min. Bound antibodies conjugated with the target proteins were detected using a chemiluminescence kit (GE Healthcare Bio-Sciences, Chicago, IL, USA). The signal intensity was measured with the LAS 4000 Mini and Multi Gauge software version 3.0 (Fujifilm).

## 3. Results

### 3.1. Genetic Findings and Clinical Features

Sequence analysis of the *PAX6* gene revealed seven heterozygous mutations, c.150_151insA (p.G65Rfs*5), c.233T>A (p.V78E), c.247G>T (p.V83F), c.383G>A (p.R128H), c.746A>T (p.Q255L), c.767T>A (p.V256E), and c.1032+5G>A. In addition, mutations c.58C>T (p.P20S) and c.1095T>G (p.N365K) were reported in our earlier study [15] (Table 3). All were novel mutations except for c.383G>A (p.R128H), which was reported in a family with mild aniridia [32]. None of the mutations was found in population databases of HGVD, Tommo3, the 1000 genomes project, or gnomAD. All corresponding codons were located at residues conserved among humans and other species, e.g., rhesus monkey, mouse, elephant, chicken, zebrafish, and frog, according to the UCSC Genome Browser. Pro20 and Gly65 were in the NTS; Val78, Val83, and Arg128 were in the CTS; Gln255 and Val256 were in the HD; and c.1032 (Gln344) and Asn365 were in the PSTD. The in silico programs predicted varying scores for pathogenicity for all variants (Table 3). Based on the ACMG criteria, all variants were taken to be pathogenic.

The associated phenotypes were FVH1 for p.P20S, p.V78E, p.V83F, p.R128H, p.Q255L, and p.N365K (Figure 1, Figure 2 and Figure 3), aniridia with corneal opacity for p.G65Rfs*5 and p.V256E, and peripheral corneal opacity with mild iris anomaly for c.1032+5G>A (Table 1). In the family with p.V256E (Family 7), the proband who had bilateral aniridia with corneal opacity and cataract also had retinal detachments after cataract or glaucoma surgery on both eyes. The mother with the same mutation had no visual symptoms, and was later found to have unilateral partial aniridia (Figure 4). She had mild foveal hypoplasia with reduced foveal pits in both eyes. Regardless of the genotype, all examined patients had bilateral foveal hypoplasia detected in the OCT images. Three patients from Family 1 (p.P20S), the mother of Family 7 (p.V256E), and Patient 17 had mild foveal hypoplasia of Grade 1b, according to the classification by Thomas et al. [33,34]. However, the other patients had severe Grade 3~4 foveal hypoplasia (Table 1). Regardless of the genotype, goniodysgenesis was observed in seven (88%) of the eight patients who underwent gonioscopy. For sixteen patients from nine families, the corrected visual acuity ranged from light perception to −0.2 logMAR units. Myopia ranging from −2.0 D to −12.5 D (median −4.3 D) was observed in sixteen of the examined patients. Nystagmus was present in five (36%) of the fourteen examined patients.

### 3.2. Functional Assays

#### 3.2.1. Expression Analysis of PAX6

The expression of PAX6 was analyzed using three types of cells, ARPE19/HPV16, LUHMES and PC3. The expression of the mRNA of *PAX6* increased LUHMES by 2.25 times when ARPE19/HPV16 was set to 1, but PC3 was not expressed (Figure 5). The PC3 cells that were transfected with Flag-PAX6 and Flag-PAX6(5a) and immunoprecipitated with Flag were analyzed by Western blotting using three kinds of commercially available antibodies (ab238527; Abcam Cambridge UK, 42-6600; Invitrogen, and sc-81649; Santa Cruz). These antibodies recognized Flag-PAX6, but the sensitivity and specificity of recognizing endogenous PAX6 was too low to assess (Appendix A). These results suggested that ARPE19/HPV16 and LUHMES expressed PAX6, but it was not determined which isoform they expressed.

#### 3.2.2. PAX6 Mutants and Transcriptional Activity via 5aCON

5aCON is a consensus site to which PAX6(5a) binds, and PAX6(5a) is known to increase the promoter activity with 5aCON. We generated different *PAX6* mutants and transfected these mutants and the wild type into 5aCON-Nluc stably transfected PC3 cells. First, we examined whether there was any change in the expression level between the wild-type PAX6 and its mutants, using a Flag antibody. Because the total number of amino acids in the PAX6 mutants, other than G65Rfs*5, was the same as that of the wild type, the signals were observed at the same molecular weight positions (Figure 6A). However, G65Rfs*5 was not observed (see Figure 6A) because its molecular weight was less than 10 kDa even with a Flag tag, due to a frameshift mutation. Because the molecular weights of PAX6 WT and G65Rfs*5 differed significantly and were expected to have different transcriptional abilities, the proteins were directly immobilized on a PVDF membrane using a slot blot apparatus. Although the signal was weak, it was possible to determine the expression ratio with the wild type (Figure 6B). Because the specificity of the anti-Flag antibody was high, we believe that the detected signal was derived from the Flag fusion protein. The results of normalizing the expression level of Flag with the expression level of β-actin are shown in Figure 6C. The expression ratio of PAX6 mutants varied greatly, and ranged from 0.6- to 2.9-fold. 

Next, we measured the Nluc activity, to evaluate the transcriptional activity of *PAX6* dependent on 5aCON. Because the Nluc activity is affected by the expression level of *PAX6*, it was normalized using the expression ratio obtained from the data shown in Figure 6C. While P20S and Q255L did not affect the transcriptional activity, all of the other mutants reduced the transcriptional activity (Figure 6D). More specifically, the mutants of CTS, V78E, V83F, and R128H, significantly decreased the transcriptional activity, as well as the truncation mutant of G65Rfs*5.

#### 3.2.3. Nuclear Translocation Analysis of PAX6 Mutants

We first examined the nuclear location of each PAX6 mutant, because some mutants were unable to activate the 5aCON-dependent transcriptional activity. PC3 cells expressing each mutant were fractionated into nuclear and cytoplasmic fractions, and the expression level of PAX6 in each fraction was determined by Western blotting (Figure 7). To evaluate the accuracy of the PVDF, we also examined the expression levels of lamin A/C and GAPDH in the nuclear and cytoplasmic fractions. As expected, lamin A/C was almost always detected in the nuclear fraction, while GAPDH was mainly detected in the cytoplasmic fraction. Based on these results, we examined whether the cytoplasmic and nuclear fractions could be effectively separated. We examined the subcellular localization of PAX6 under these fractionation conditions. We found that most of the G65Rfs*5 mutants remained in the cytoplasm and did not translocate into the nucleus, while the other mutants had translocated into the nucleus.

#### 3.2.4. Binding Analysis of PAX6 Mutants and 5aCON

The pull-down analysis adopted in this analysis is a method of mixing the extracted nuclear proteins and oligonucleotides, recovering the oligonucleotides with magnetic beads, and detecting the proteins bound thereto by Western blotting. This method enables quantitative analysis of proteins bound to oligonucleotides. On the other hand, the traditional gel shift assay similarly mixes extracted nuclear proteins and radiolabeled oligonucleotides, electrophoreses the mixture, and analyzes the signals with a delayed migration signal. The delayed migration signal contains bound oligonucleotides and proteins, but it is not known which proteins among the extracted nuclear proteins are bound. Therefore, it is necessary to add a specific antibody, in this case the anti-Flag antibody, against protein, and perform the supershift assay. However, when the amount of added antibody is small, it is not possible to supershift all the signals bound by the target protein. Therefore, quantitative analysis is difficult with the supershift assay, and it is considered to be a qualitative analysis. For these reasons, we adopted a pull-down analysis that allows quantitative analysis and did not use radiation.

Because we found that mutants other than G65Rfs*5 could translocate into the nucleus, we next examined whether these mutants could bind to 5aCON. We mixed an oligonucleotide containing the sequence of 5aCON immobilized on magnetic beads with nuclear protein extracted from the nucleus of each cell, and analyzed the PAX6 protein bound to 5aCON by Western blotting. P20S and Q255L, which had normal transcriptional activity, bound to 5aCON as much as, or even more than, the wild-type PAX6 (Figure 8A). In addition, V78E, V83F, R128H, and V256E, which have low 5aCON transcriptional activity, had low binding to 5aCON. On the other hand, N365K had a 50% reduction in the transcriptional activity of 5aCON, but bound more to 5aCON than to the wild-type PAX6.

#### 3.2.5. Transcription Activity Analysis Using Gal4 System

Although N365K was able to bind to 5aCON, its 5aCON-mediated luciferase activity was low. The GAL4 one hybrid transcription system was used to determine the reason for this. The GAL4 protein is a transcription factor that binds to a specific DNA sequence known as the upstream activating sequence (UAS), which is located upstream from the target gene. The binding of GAL4 to UAS recruits other transcription factors including the general transcription factors such as the TATA-binding protein (TBP) and RNA polymerase II, to form a pre-initiation complex that initiates transcription of the target gene. Luciferase activity is affected by the DNA-binding and transcriptional activities of transcription factors. That is, if the mutants have different DNA-binding abilities, luciferase activity cannot be used to evaluate transcriptional activity. Therefore, using the GAL4 one hybrid system, in which the DNA-binding ability is equivalent, we compared the transcription activity of the wild type and the mutant. In this assay, only the PSTD region, a putative transactivation domain, was fused to GAL4. The protein expression levels from transfection were quantified (Figure 8B), and the Nluc activity was normalized for protein abundance (Figure 8C). The results showed that the PSTD transcriptional activity of N365K was found to be 32% of that of the wild type.

## 4. Discussion

We analyzed patients with not only FVH1, but with foveal hypoplasia associated with varying degrees of corneal opacities and aniridia, viz., Family 2 with p.G65Rfs*5, Family 7 with p.V256E, and Family 8 with c.1032+5G>A. In these three families, the responsible mutations were at loci other than the CTS. Interestingly, Family 7 showed remarkable phenotypic variations, i.e., the proband had typical aniridia with severe corneal opacity, whereas the mother had unilateral partial aniridia and bilateral foveal hypoplasia with preserved BCVA. The functional assay showed that the truncation p.G65Rfs*5 mutation had no transcriptional activity. The transcript of the truncation mutation could possibly be degraded by nonsense-mediated decay. This was supported by the phenotype of the patient, which could be caused by haploinsufficiency of the *PAX6* gene. Nonetheless, when translated, the protein would not function because it did not translocate into the nucleus. p.V256E had low 5aCON transcriptional activity. We did not analyze the c.1032+5G>A, because the putative transcript of this splicing mutation was uncertain. However, the mutant can be transcribed as one of other minor transcripts, e.g., p.S346N for NM_001310159, and further studies are needed to determine the effects of this mutation on the formation of the fovea.

The remaining families had a consistent FVH1 phenotype, all of which had missense mutations. However, the severity of the foveal hypoplasia differed among the families. In Families 3, 4, and 5, their responsible mutations, p.V78E, p.V83F and p.R128H, were in the CTS, and they consistently showed severe foveal hypoplasia of Grade 3 or 4. Earlier studies showed that most of the mutations causing FVH1 were found in the CTS [12,13]. From our results on the mutations other than the CTS, the clinical severity was divided into mild foveal hypoplasia (p.P20S and p.N365K) [15], and severe foveal hypoplasia (p.Q255L). Our results showed that the functional impairments associated with the mutations were diverse. The mutants P20S in the NTS and Q255L in the HD did not affect the transcriptional activity, suggesting a different action from mutants in the CTS that induce hyperactivation of the NTS [6]. Mutant N365K can bind more to the 5aCON than the wild-type *PAX6*; however, it had a reduced transcriptional activity. To determine the reason for this discrepancy, we used the GAL4 transcription system in the PSTD, which is known to have transactivation actions [35]. We found that the transactivation of PSTD by N365K was reduced to 32%. This supports the idea that the transactivation by PAX6 was dependent on the location of the mutation and type of the DNA-binding site [36].

There are limitations in this study. An important limitation was that only the mutations we found were evaluated, and the number of patients with the mutations was small. However, isolated foveal hypoplasia is a rare condition, and only a few mutations have been analyzed functionally. Thus, there is limited information on the characteristics of foveal hypoplasia. The other limitations include lack of information of genes directly and indirectly dysregulated by PAX6 and PAX6(5a) in human fovea. We cannot explain why the phenotypically significant mutants of P20S and Q255L showed a comparable level for the expression, nuclear translocation, transcriptional activity, and binding to 5aCON, to the wild-type PAX6.

## 5. Conclusions

The results showed that mutations associated with foveal hypoplasia underlie a functional divergence from the DNA-binding ability to the transcriptional activity. Mutations in the CTS consistently cause severe foveal hypoplasia. Widely distributed mutations in the *PAX6* gene, not limited to the CST region, were responsible for isolated foveal hypoplasia. Our study expands the mutation spectrum of foveal hypoplasia, which will be helpful for genetic counseling and provides information that helps in the understanding of the fovea.

## Figures and Tables

**Figure 1 genes-14-01483-f001:**
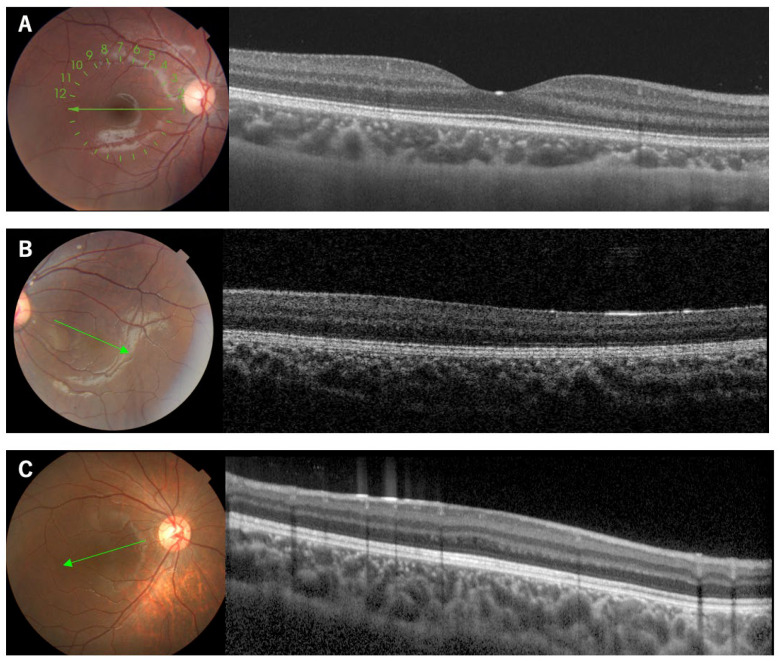
Fundus photographs and OCT images of a normal subject and two patients with isolated foveal hypoplasia. The OCT images are from line scans of the fundus photographs designated by the corresponding lines. Note that a normal subject shows a foveal pit (**A**), while the patients lack a foveal pit ((**B**): Patient 8, (**C**): Patient 14).

**Figure 2 genes-14-01483-f002:**
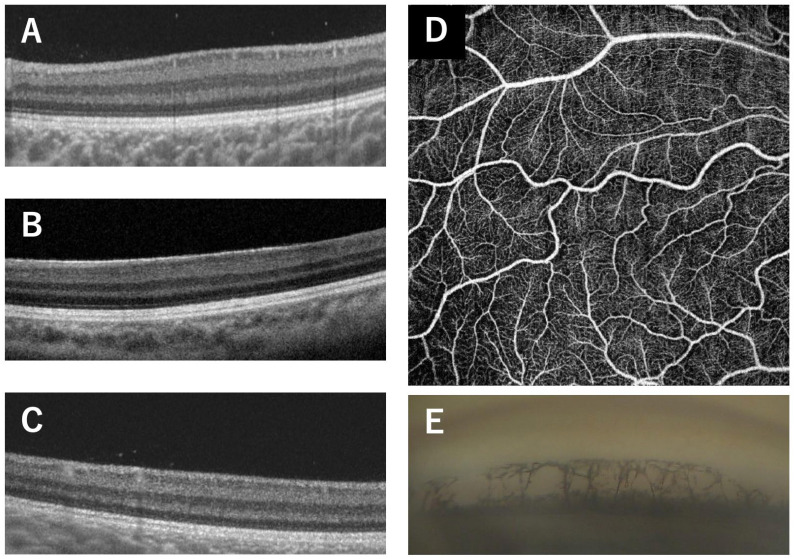
Clinical findings of Family 2 (Patients 5, 6, 7) carrying the p.V78E mutation in the *PAX6* gene. OCT images of all patients show no foveal pit ((**A**), Patient 5; (**B**), Patient 6; (**C**), Patient 7). The 1-year-old brother (Patient 7) underwent examinations under general anesthesia which showed an absence of the foveal avascular zone in the superficial layers of the OCTA image (**D**) and goniodysgenesis (**E**). The OCT scan images were taken at a level across the presumed foveal region.

**Figure 3 genes-14-01483-f003:**
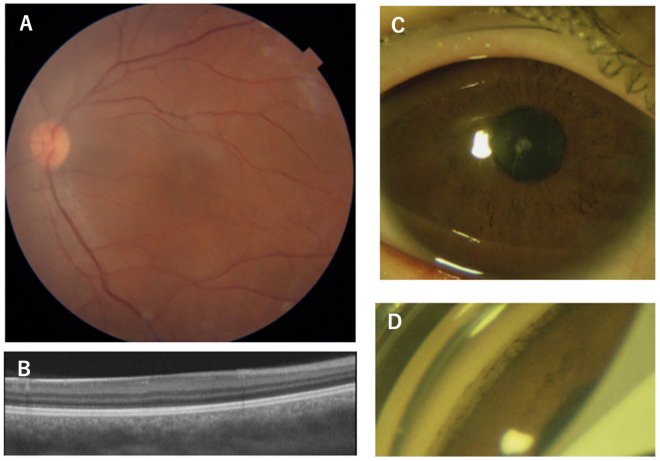
Clinical findings of Patient 11, the 21-year-old male patient with the p.Q255L mutation in the *PAX6* gene. Fundus photographs shows normal retinal appearance (**A**). OCT image shows no foveal pit (**B**). Slit-lamp photograph shows no obvious changes in the iris surface, and slight cataract (**C**). Gonioscopic photograph shows goniodysgenesis (**D**). The OCT scan image was taken at a level across the presumed foveal region.

**Figure 4 genes-14-01483-f004:**
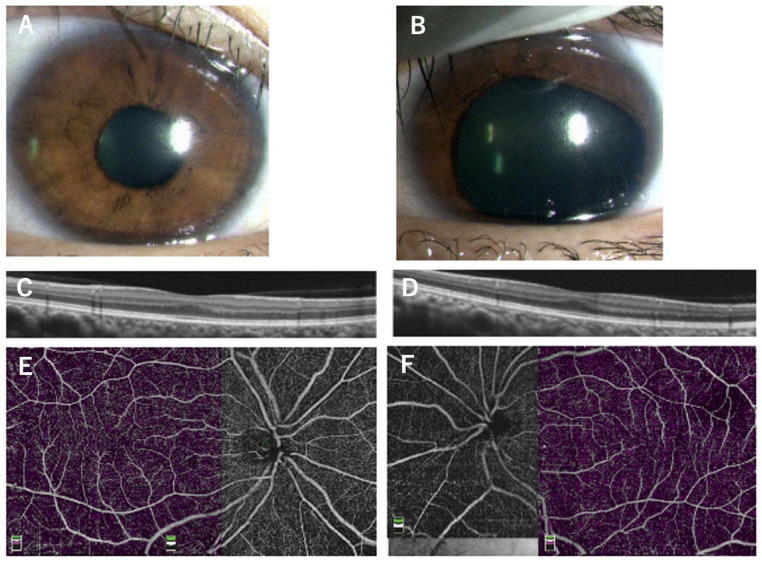
Images of a 36-year-old mother of an aniridia patient with p.V256E mutation in the *PAX6* gene. The images of the iris of the right eye are normal (**A**) but with partial aniridia in the left eye (**B**). OCT and the superficial layers of OCTA images show reduced foveal pit (**C**,**D**) and no foveal avascular zone (**E**,**F**). The OCT scan images were taken at a level across the presumed foveal region.

**Figure 5 genes-14-01483-f005:**
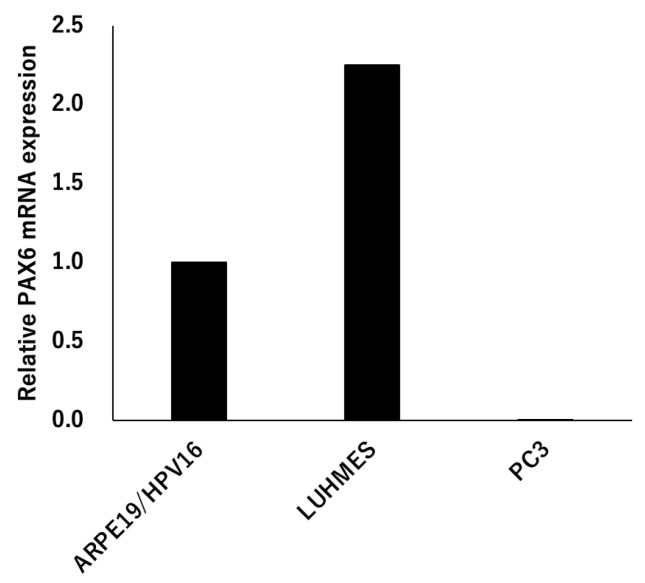
*PAX6* expression. Total RNA was extracted from ARPE19/HPV16, LUHMES, and PC3 cells, and the expression level of *PAX6* mRNA was analyzed by qRT-PCR (ΔΔCT method). The expression level of ARPE19/HPV16 was set to 1 (*n* = 1).

**Figure 6 genes-14-01483-f006:**
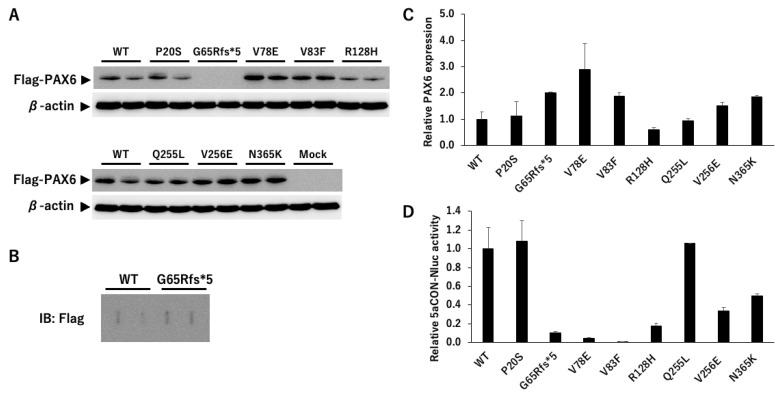
Expression of exogenous PAX6. (**A**) Flag-tagged wild-type PAX6 and its mutants were transfected into PC3-5aNluc cells, and the cells were harvested after 48 h. Cell lysates were examined by SDS-PAGE, and the membrane was blotted with anti-Flag antibody to detect the expression of the transfected PAX6 constructs. An anti-β-actin antibody was used as a loading control to normalize the protein expression levels (*n* = 2). (**B**) Cell lysates expressing wild-type PAX6 and G65Rfs*5 were blotted onto membranes using slot blotting, and then blotted with anti-Flag antibody (*n* = 2). (**C**) The results from A and B were normalized using the common signal which was the wild-type PAX6. The expression ratio of Flag to β-actin was calculated, and the value of the wild type was set to 1 (*n* = 2). (**D**) 5aCON-Nluc activity induced by each PAX6 protein was measured, and normalized by β-actin expression levels in the cells (**A**,**B**).

**Figure 7 genes-14-01483-f007:**
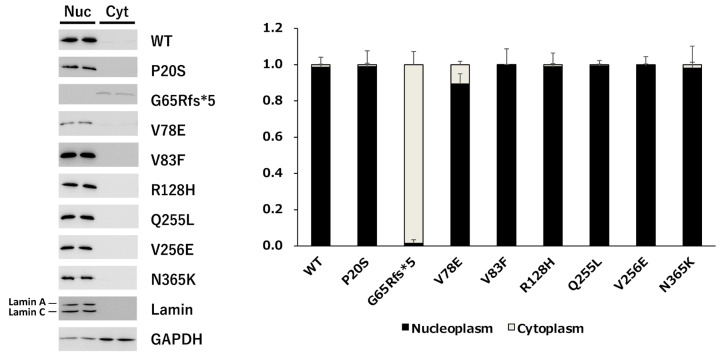
Subcellular localization of wild-type PAX6 and its mutants. Plasmids expressing the wild-type PAX6 and its mutants were transfected into PC3 cells, and the cells were harvested. The harvested cells were fractionated into nucleoplasm and cytoplasm, and lysed in equal amounts. Equal volumes of the cell lysate were aliquoted, subjected to SDS-PAGE, and then transferred onto a membrane. The membrane was probed with anti-Flag antibody to detect the expression of the transfected PAX6 constructs. Lamin and GAPDH were also blotted as loading controls to confirm the accuracy of the fractionation procedure. The total of each expression level was set to 1, and the ratio of nucleoplasm to cytoplasm was determined (*n* = 2). Nuc and Cyt indicate nucleoplasm and cytoplasm, respectively.

**Figure 8 genes-14-01483-f008:**
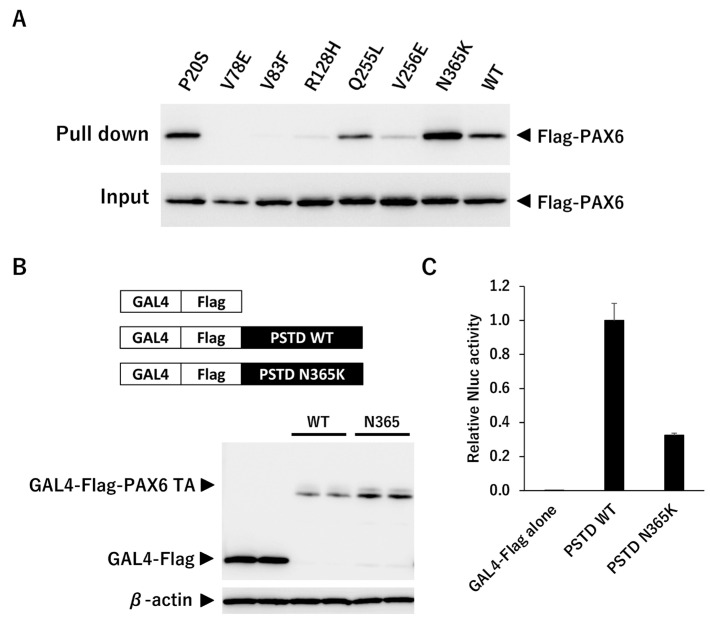
5aCON binding activity and PSTD transcriptional activity. (**A**) The upper panel shows the results of pulling down the indicated wild-type PAX6 and mutant PAX6 with immobilizing 5aCON on magnetic beads. The expression level of each Flag-PAX6 in the cell lysate used for the pull-down is shown in the lower panel. The numbers at the bottom of the panel indicate each signal intensity ratio when WT is set to 1. (**B**) PC3-9Gal4Nluc cells were transfected with GAL4-Flag-tagged PTSD of the wild type or N365K, and harvested after 48 h. Cell lysates were separated by SDS-PAGE and transferred to a membrane for Western blotting using an anti-Flag antibody to detect the expression of each construct. β-actin was used as a loading control to normalize protein expression levels. (*n* = 2). (**C**) 9xGAL4UAS-Nluc activation by PSTD WT and N365K was measured and normalized byβ-actin expression levels in cells (**B**).

**Table 1 genes-14-01483-t001:** Mutations in the *PAX6* gene and clinical features in patients with foveal hypoplasia.

Patient	Age (Years), Sex	Family No.	Diagnosis	Mutation (NM_000280.4)	PAX6 Domain	logMAR Visual Acuity (R/L)	Refractive Error (R/L)	Nystagmus	Gonio Dysgenesis	OCT Grade
1 *	1325, F	1 (proband)	FVH1	c.58C>T (p.P20S)	PD (NTS)	0.5/0.2	−3.75/−4.0	present	present	1b
2 *	34, F	1 (mother)	FVH1	c.58C>T (p.P20S)	PD (NTS)	0.1/0.1	−0.5/0	absent	present	1b
3 *	64, M	1 (grandfather)	FVH1	c.58C>T (p.P20S)	PD (NTS)	0.1/0	NA	absent	present	1b
4	55, M	2 (proband)	aniridia, corneal opacity	c.150_151insA (p.G65Rfs*5)	PD (NTS)	NA	NA	NA	NA	NA
5	3, F	3 (proband)	FVH1	c.233T>A (p.V78E)	PD (CTS)	1.2/1.3	−2.0/−2.0	present	NA	4
6	33, F	3 (mother)	FVH1	c.233T>A (p.V78E)	PD (CTS)	1.0/1.0	−10.25/−12.0	present	present	4
7	1, M	3 (brother)	FVH1	c.233T>A (p.V78E)	PD (CTS)	1.2/1.4	+3.0/+3.75	present	present	4
8	6, M	4 (proband)	FVH1	c.247G>T (p.V83F)	PD (CTS)	0.5/0.4	−7.125/−7.25	absent	NA	4
9	10, M	4 (brother)	FVH1	c.247G>T (p.V83F)	PD (CTS)	0/0.1	−4.5/−3.875	absent	NA	4
10	68, M	5 (proband)	FVH1	c.383G>A (p.R128H)	PD (CTS)	0.2/0.3	NA	NA	NA	3
11	21, M	6 (proband)	FVH1	c.764A>T (p.Q255L)	HD	0.3/0.4	+0.75/−1.0	present	present	4
12	8, F	7 (proband)	aniridia, corneal opacity, secondary retinal detachments	c.767T>A (p.V256E)	HD	LP/1.7	NA/+16.5	NA	NA	NA
13	36, F	7 (mother)	unilateral partial aniridia	c.767T>A (p.V256E)	HD	NA	NA	absent	absent	1b
14	6, M	8 (proband)	peripheral corneal opacity, mild iris anomaly, exotropia	c.1032+5G>A	PSTD	0.1/0.1	+0.25/+0.5	absent	NA	3
15	41, F	8 (mother)	peripheral corneal opacity, mild iris anomaly	c.1032+5G>A	PSTD	0.2/0.1	+2.0/−1.25	absent	NA	3
16	37, M	8 (uncle)	peripheral corneal opacity, mild iris anomaly	c.1032+5G>A	PSTD	0.6/0.5	−2.0/−4.0	absent	NA	3
17 *	6, F	9 (proband)	FVH1	c.1095T>G (p.N365K)	PSTD	−0.2/−0.1	−5.5/−5.75	absent	present	1b

* Patients 1, 2, 3, and 17 have been reported in reference [15]. CTS, C-terminal subdomain; F, female; FVH1, autosomal dominant isolated foveal hypoplasia; HD, homeodomain; M, male; NA, not available; NTS, N-terminal subdomain; PD, paired domain; PSTD, proline-serine-threonine-rich domain. The visual acuity was measured with the Teller acuity cards for Patients 5 and 7.

**Table 2 genes-14-01483-t002:** Oligonucleotides.

Primer Name	Sequence (5’ – 3’)
PAX6 P20S FW	GGGCGGTCACTGCCGGACTCCACCC
PAX6 P20S RV	GCAGTGACCGCCCGTTGACAAAGAC
PAX6 V78E FW	CGAGAGAAGCGACTCCAGAAGTTGT
PAX6 V78E RV	GTCGCTTCTCTCGGTTTACTACCAC
PAX6 V83F FW	CCAGAATTTGTAAGCAAAATAGCCC
PAX6 V83F RV	TTACAAATTCTGGAGTCGCTACTCT
PAX6 R128H FW	TTCTTCACAACCTGGCTAGCGAAAA
PAX6 R128H RV	AGGTTGTGAAGAACTCTGTTTATTG
PAX6 Q255L FW	GAATACTGGTATGGTTTTCTAATCG
PAX6 Q255L RV	CATACCAGTATTCTTGCTTCAGGTA
PAX6 V256E FW	TACAGGAATGGTTTTCTAATCGAAG
PAX6 V256E RV	AACCATTCCTGTATTCTTGCTTCAG
PAX6 N365K FW	GGTGAAGGGGCGGAGTTATGATACC
PAX6 N365K RV	CCGCCCCTTCACCGAAGGGCTGGTG
5aCON FW 5-biotin	(Biothin) ATCTGAACATGCTCAGTGAATGTTCATTGACTCTC
5aCON RV	GAGAGTCAATGAACATTCACTGAGCATGTTCAGAT
TA FW	(EcoRI) gaattcGAAAAACTGAGGAATCAGAGAAG

**Table 3 genes-14-01483-t003:** Pathogenicity assessment of the mutations in the *PAX6* gene.

Nucleotide Change	Amino Acid Change	Polyphen2 HumDIV [13] (cutoff = 0.85)	GERP++ [14] (cutoff = 2)	REVEL [15] (cutoff = 0.5) *	M-CAP [16] (cutoff = 0.025)	CADD Phred [17] (cutoff = 15) **	SIFT [18] (cutoff = 0.05)	Pathogenicity (Evidenced Criteria Points) ***
c.58C>T	p.P20S	0.980	4.48	0.927	0.939	23.900	0.010	Pathogenic (PS = 1, PM = 2, PP = 4)
c.150_151insA	p.G65Rfs*5							Pathogenic (PVS = 1, PS = 1, PM = 2, PP = 1)
c.233T>A	p.V78E	1.000	5.35	0.979	0.957	28.600	0.000	Pathogenic (PS = 1, PM = 3, PP = 4)
c.247G>T	p.V83F	1.000	5.35	0.942	0.916	29.800	0.000	Pathogenic (PS = 1, PM = 2, PP = 4)
c.383G>A	p.R128H	0.997	3.83	0.933	0.786	28.100	0.000	Pathogenic (PS = 2, PM = 3, PP = 3)
c.764A>T	p.Q255L	0.999	5.47	0.984	0.822	34.000	0.000	Pathogenic (PS = 1, PM = 2, PP = 3)
c.767T>A	p.V256E	1.000	5.53	0.976	0.824	32.000	0.000	Pathogenic (PS = 2, PM = 3, PP = 3)
c.1032+5G>A	-							Pathogenic (PVS = 1, PM = 2, PP = 2)
c.1095T>G	p.N365K	0.736	NA	0.499	0.145	8.266	0.020	Pathogenic (PS = 2, PM = 2, PP = 2)

Underlined values are “pathogenic” according to the cutoff values (refs. [22,23,24,25,26,27]). NA: not available. * 75.4% of disease mutations but 10.9% of neutral variants, ** ≤1% percentile highest scores, *** based on the reference [28]: PVS, very strong; PS, strong; PM, moderate; PP, supporting.

## Data Availability

Data are available on request.

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
