# Peer review of "Functional Characteristics of Diverse PAX6 Mutations Associated with Isolated Foveal Hypoplasia"

_genes, 2023, doi:10.3390/genes14071483_

Round 1
Reviewer 1 Report
This is a multicenter retrospective case series study aimed to characterize 17 patients from 9 Japanese families with autosomal dominant isolated foveal hypoplasia (FVH1) and the other PAX6-related phenotypes with foveal hypoplasia. The authors showed the clinical features of both anterior and posterior segment of eyes. They also did sophisticated experiment to investigate the PAX6 mutants in the aspect of protein expression, transcriptional activity via 5aCON, nuclear translocation, binding of 5aCON, and transcription activity analysis using Gal4 system.
Several comments that may help to improve this manuscript.
1. Line 90: “OCT images were recorded from all patients”. However, the items of clinical images are not complete for all the families. The characterization of foveal hypoplasia is dependent on the precise scanning through the foveal pit; therefore, it would be better to provide a small reference enface image showing the scanning line for Figure 1 to 3.
2. OCTA showing no foveal avascular zone is good supportive evidence. The retinal layer in which blood flow signal was obtained should be specified. The details, such as the type and manufacturer of the machine of OCTA used in clinical examination was missed in the Method.
3. Table 1 shows data of visual acuity and OCT grading of pediatric patients of 1 and 3 years old (patient 7 and 5). It is curious how was the visual acuity measured and how to acquire accurate OCT image for young kids in this study.
4. The use of cell lines was complicated. In the Figure 4, the purpose of study seems to be proving that PC3 cells do not express PAX6. It is interesting to know the reasons of choosing APRE19/HPV16 and LUHMES as positive controls. The role of these two cell lines is not clear in the other part of this study. Meanwhile, COS1 cells, which was used in production of lentivirus carrying plasmids, was not mentioned in 2.3.1. Cell lines and culture.
5. Line 173: “A total of 1x105 PC3-5aNluc, PC3-9Gal4Nluc, and PC3 cells were seeded into a 35 mm dish” The writing of 1x105 is supposed to be 1x105. Please be careful of the superscript. In addition, does “PC-5aNluc” represent “pCDH 5aCON-Nluc G2P” plasmid?
6. Line 293: The description “5aCON-Nluc (nanoluciferase) was stably transfected into PC3 cells not expressing PAX6 (PC3-5aNluc). 5aCON is a consensus site to which PAX6(5a) binds, and PAX6(5a) is known to increase the promoter activity with 5aCON.” is better placed to the Method 2.3.3. Plasmid preparation to clarify the purpose of functional study.
7. Line 281: “The PC3 cells that were transfected with Flag-PAX6 and Flag-PAX6(5a) and immunoprecipitated with Flag were analyzed by Western blotting using two kinds of commercially available antibodies” What is the antibody exactly is not mentioned in the method.
8. It is not clear about the exact meaning of this sentence in line 380 “As a feature of the GAL4 system, even if the DNA-binding ability of each transcription factor is originally different, it can be regarded as the same by fusing GAL4, and therefore it is necessary to remove the original DNA binding region.”.
9. Figure 7 A. Quantification of the pulling down indicated wild-type PAX6 and mutant PAX6 with immobilizing 5aCON on 365 magnetic beads is suggested.
10. How to explain the loss of function for PAX6 with P20S and Q255L mutants, which had normal expression level, normal nuclear translocation, normal transcriptional activity, and normal bound to 5aCON when compared to WT?
Author Response
POINT-BY-POINT RESPONSE
Reviewer 1
Response: Thank you so much for your deliberate review. Your suggestions make our manuscript clearer. We have answered all your comments point-by-point.
This is a multicenter retrospective case series study aimed to characterize 17 patients from 9 Japanese families with autosomal dominant isolated foveal hypoplasia (FVH1) and the other PAX6-related phenotypes with foveal hypoplasia. The authors showed the clinical features of both anterior and posterior segment of eyes. They also did sophisticated experiment to investigate the PAX6 mutants in the aspect of protein expression, transcriptional activity via 5aCON, nuclear translocation, binding of 5aCON, and transcription activity analysis using Gal4 system.
Several comments that may help to improve this manuscript.
- Line 90: “OCT images were recorded from all patients”. However, the items of clinical images are not complete for all the families. The characterization of foveal hypoplasia is dependent on the precise scanning through the foveal pit; therefore, it would be better to provide a small reference enface image showing the scanning line for Figure 1 to 3.
Answer: Thank you for the comment. Additional images from other families (available) were added as Figure 1 except for Families 1 and 9 for which their images have been published earlier (ref. 15). In Figure1, the representative OCT images of both patients and normal individual were shown in the corresponding fundus photographs where the reference lines were drawn. For Figure 2 to 4, the explanation was added instead of reference fundus images: 298-299, 304-305, 309-310
- OCTA showing no foveal avascular zone is good supportive evidence. The retinal layer in which blood flow signal was obtained should be specified. The details, such as the type and manufacturer of the machine of OCTA used in clinical examination was missed in the Method.
Answer: The details were added in the text: 99-102, 298, 309
- Table 1 shows data of visual acuity and OCT grading of pediatric patients of 1 and 3 years old (patient 7 and 5). It is curious how was the visual acuity measured and how to acquire accurate OCT image for young kids in this study.
Answer: Visual acuity was measured using the Teller acuity cards for Patient 5 (3-year-old) and Patient 7 (1-year-old): added in the foot note of Table 1.
Compared with OCTA images, OCT images were available from some selected young children even at the age of 1-year-old (actually nearly 2-year-old) because they can be obtained at shorter scanning time.
- The use of cell lines was complicated. In the Figure 4, the purpose of study seems to be proving that PC3 cells do not express PAX6. It is interesting to know the reasons of choosing APRE19/HPV16 and LUHMES as positive controls. The role of these two cell lines is not clear in the other part of this study. Meanwhile, COS1 cells, which was used in production of lentivirus carrying plasmids, was not mentioned in 2.3.1. Cell lines and culture.
Answer: As the reviewer speculated, LUHMES and ARPE19 cells were used as positive control against PC3 cells for PAX6 expression. In our microarray analysis, LUHMES and ARPE19 cells expressed higher PAX6 mRNA expression than other normal immortalized human cells (pleural mesothelial cells, ovarian epithelial cells, and lung fibroblasts). PAX6 is also known to function in retina and neurons, so it is not surprising that PAX6 is expressed in LUHMES and ARPE19 cells. In fact, PAX6 mRNA expression was confirmed in both cells. In addition, information on COS1 cells has been added to "2.3.1. Cell lines and culture":124-132.
- Line 173: “A total of 1x105 PC3-5aNluc, PC3-9Gal4Nluc, and PC3 cells were seeded into a 35 mm dish” The writing of 1x105 is supposed to be 1x105. Please be careful of the superscript. In addition, does “PC-5aNluc” represent “pCDH 5aCON-Nluc G2P” plasmid?
Answer: Corrected the notation of the number of cells. "PC3-5aNluc" is the name of stable expression cells, not "pCDH 5aCON-Nluc G2P" plasmid. This was explained at the end of the First Edition section 2.3.4.
- Line 293: The description “5aCON-Nluc (nanoluciferase) was stably transfected into PC3 cells not expressing PAX6 (PC3-5aNluc). 5aCON is a consensus site to which PAX6(5a) binds, and PAX6(5a) is known to increase the promoter activity with 5aCON.” is better placed to the Method 2.3.3. Plasmid preparation to clarify the purpose of functional study.
Answer: A words “not expressing PAX6” was added in Methods 2.3.4: 183. The sentence “5aCON-Nluc (nanoluciferase) was stably transfected into PC3 cells not expressing PAX6 (PC3-5aNluc)” was removed to avoid confusion in 3.2.2. An explanatory paragraph was added in 3.2.4: 385-398.
- Line 281: “The PC3 cells that were transfected with Flag-PAX6 and Flag-PAX6(5a) and immunoprecipitated with Flag were analyzed by Western blotting using two kinds of commercially available antibodies” What is the antibody exactly is not mentioned in the method.
Answer: Added as suggested: 316-317 (to be more precise, an additional antibody was used for the confirmation: 315).
- It is not clear about the exact meaning of this sentence in line 380 “As a feature of the GAL4 system, even if the DNA-binding ability of each transcription factor is originally different, it can be regarded as the same by fusing GAL4, and therefore it is necessary to remove the original DNA binding region.”.
Answer: The sentence was changed for better explanation: 428-432. In relation, the sentence in 2.3.3 was modified to add “excluding the DNA binding region”: 170-171
- Figure 7 A. Quantification of the pulling down indicated wild-type PAX6 and mutant PAX6 with immobilizing 5aCON on 365 magnetic beads is suggested.
Answer: We added the signal intensity ratio in the Figure 7A with the explanation in the legend.
- How to explain the loss of function for PAX6 with P20S and Q255L mutants, which had normal expression level, normal nuclear translocation, normal transcriptional activity, and normal bound to 5aCON when compared to WT?
Answer: Thank you for the point. The comment was incorporated in the limitation: 475-478. We speculate that since this analysis was performed with PC3 cells that do not express PAX6, a different reaction from cells expressing PAX6 may have occurred. For example, in the case of forming a complex with PAX6 to increase transcriptional ability, if PAX6 mutation reduces the ability to form a complex, there may be functional differences between the wild type and P20S and Q255L. To confirm this, it is necessary to knock out the PAX6 gene of ARPE19/HPV16 and LUHMES and perform the same experiment as this time. However, because these are so speculative, they are not commented in the text.

Reviewer 2 Report
The manuscript by Itsuka Matsushita and co-workers is focused on the identification of disease-causing variants in PAX6 gene linked to isolated foveal hypoplasia in 17 patients followed by studies to understand molecular mechanisms of individual mutations. The Introduction is comprised from five paragraphs, however, two of them are extremely short. Additional background information is needed (see below). The first portion of the manuscript (Tables 1 and 2, Figures 1-3) is well described. The authors are encouraged to show additional images of individual patients and normal macula, showing the foveal pit. The transition to molecular biology experiments needs much better justification as not all assays reported in earlier publications are feasible to conduct within a single manuscript including eight PAX6 mutations. The reporter assays are solely based on using a tandem of Pax6(5a)-binding sites identified by Epstein et al. 1994 [ref 7] that bind recombinant GST-PD(5a) proteins lacking the HD and PSTD using the PC3 cells lacking endogenous expression of PAX6 mRNAs (Fig. 4). Next, reporter assays were conducted using co-transfections with the canonical FLAG-PAX6 wild type and mutated expression vectors in stable cells to simplify the normalizations. Earlier versions of the 5aCON-Nluc reporter were used by others in transient co-transfections using both PAX6, PAX6(5a), or both of them at ratios reflecting their expression in different cell types. As expected, individual PAX6 mutants caused no changes, moderate or major reductions of the reporters (Fig. 5). Compared to the majority of earlier studies, the authors also evaluated possible changes in subcellular localization of PAX6 (Fig. 6) and found mostly cytoplasmic localization of PAX6 G65Rfs*5 frameshift mutant as this protein is just a severely truncated paired domain (PD). They further linked it to nonsense mediated RNA decay. The DNA-binding properties of this series of PAX6 mutants were examined using DNA-bound to magnetic beads and nuclear extracts and quantified using western blots (Fig. 7A). Finally, The PAX6 N365K missense mutant and normal control were examined using Gal4 system suitable to evaluate activities of transcriptional activation domain (Figs. 7BC). Taken together, this manuscript provides novel data on foveal abnormalities in human patients caused by mutations in PAX6 protein coding regions.
Additional comments:
1) Abstract and Introduction: Replace abbreviations NTS and CTS with commonly used PAI and RED-subdomains.
2) Abstract: Space permitting, insert a sentence what is the fovea. Likewise, state that PAX6 is a sequence-specific DNA-binding transcription factor …
3) Abstract: Space permitting, state explicitly that a single reporter containing a representative PAX6 binding site was used.
4) Introduction (line 40): Insert additional information on human fovea (see PMID: 23500068 and other relevant reviews).
5) Introduction (line 45): Ref [2] is a review on a spectrum of human eye diseases caused by mutations in PAX6. For master regulator, vertebrate and invertebrate systems other references include but are not limited to PMIDs: 10461206, 12100888, and 18331895.
6) Introduction (line 47): Move current ref [3] after ….corneal epithelium [3]. Replace reference [3] in the second sentence here …in the central nervous system [e.g. PMID: 35682795 and 33515537].
7) Introduction (line 48-49): Merge this very short paragraph with the next one to much better define structure of PAX6 and PAX6(5a) and differences in their DNA-binding mechanisms.
8) Introduction (line 58): Current ref [7] only talks about GST-PD(5a), studies of PD, HD, PD/HD and PD95a)/HD.
9) Reformate the final manuscript to prevent splitting Table 1 into two pages. It is also possible to eliminate the final column (Reference) and state ref [11] within the heading.
10)M&M (line 114): You may consider this link as well: https://clinicalgenome.org/affiliation/40072/
11)M&M (line 133): State explicitly that the reporter contains a single copy of DNA sequence from ref [7]
12)Reformate the final manuscript to prevent splitting Figure 2 into two pages.
13)Subsection 3.2.2.: Explain in detail that a wide range of synthetic and natural reporters were used in similar studies but that there is no known direct target of PAX6 that is expressed in retinal cells forming the fovea. Since the 5aCON-luc is activated by both PAX6 and PAX6(5a), it can be used for the purpose of the present study.
14)Subsection 3.2.4.: Explain the current method and compare with more traditional gel shift assay.
15)Discussion (last paragraph): Other limitations include lack of knowledge of genes directly and indirectly dysregulated by PAX6 and PAX6(5a) in human fovea.
16)Provide list of Abbreviations.
Author Response
POINT-BY-POINT RESPONSE
Reviewer 2
Response: Thank you so much for your deliberate review. Your suggestions make our manuscript clearer. We have answered all your comments point-by-point.
Comments and Suggestions for Authors
The manuscript by Itsuka Matsushita and co-workers is focused on the identification of disease-causing variants in PAX6 gene linked to isolated foveal hypoplasia in 17 patients followed by studies to understand molecular mechanisms of individual mutations. The Introduction is comprised from five paragraphs, however, two of them are extremely short. Additional background information is needed (see below). The first portion of the manuscript (Tables 1 and 2, Figures 1-3) is well described. The authors are encouraged to show additional images of individual patients and normal macula, showing the foveal pit. The transition to molecular biology experiments needs much better justification as not all assays reported in earlier publications are feasible to conduct within a single manuscript including eight PAX6 mutations. The reporter assays are solely based on using a tandem of Pax6(5a)-binding sites identified by Epstein et al. 1994 [ref 7] that bind recombinant GST-PD(5a) proteins lacking the HD and PSTD using the PC3 cells lacking endogenous expression of PAX6 mRNAs (Fig. 4). Next, reporter assays were conducted using co-transfections with the canonical FLAG-PAX6 wild type and mutated expression vectors in stable cells to simplify the normalizations. Earlier versions of the 5aCON-Nluc reporter were used by others in transient co-transfections using both PAX6, PAX6(5a), or both of them at ratios reflecting their expression in different cell types. As expected, individual PAX6 mutants caused no changes, moderate or major reductions of the reporters (Fig. 5). Compared to the majority of earlier studies, the authors also evaluated possible changes in subcellular localization of PAX6 (Fig. 6) and found mostly cytoplasmic localization of PAX6 G65Rfs*5 frameshift mutant as this protein is just a severely truncated paired domain (PD). They further linked it to nonsense mediated RNA decay. The DNA-binding properties of this series of PAX6 mutants were examined using DNA-bound to magnetic beads and nuclear extracts and quantified using western blots (Fig. 7A). Finally, The PAX6 N365K missense mutant and normal control were examined using Gal4 system suitable to evaluate activities of transcriptional activation domain (Figs. 7BC). Taken together, this manuscript provides novel data on foveal abnormalities in human patients caused by mutations in PAX6 protein coding regions.
Answer: Thank you for the comments. Some short paragraphs were modified to combine with the adjacent paragraphs.
Additional comments:
1) Abstract and Introduction: Replace abbreviations NTS and CTS with commonly used PAI and RED-subdomains.
Answer: Thank you. We left the NTS and CTS as they are because many papers including a very recent review in the Genes journal (ref. 10) still uses the NTS and CTS as well as the established human genetic and disease database (OMIM, https://www.omim.org/).
2) Abstract: Space permitting, insert a sentence what is the fovea. Likewise, state that PAX6 is a sequence-specific DNA-binding transcription factor …
3) Abstract: Space permitting, state explicitly that a single reporter containing a representative PAX6 binding site was used.
Answer: The abstract was revised by incorporating the suggestion (highlighted in Red).
4) Introduction (line 40): Insert additional information on human fovea (see PMID: 23500068 and other relevant reviews).
Answer: Revised as suggested: 43-44
5) Introduction (line 45): Ref [2] is a review on a spectrum of human eye diseases caused by mutations in PAX6. For master regulator, vertebrate and invertebrate systems other references include but are not limited to PMIDs: 10461206, 12100888, and 18331895.
Answer: The reference was replaced with [3]: 51
6) Introduction (line 47): Move current ref [3] after ….corneal epithelium [3]. Replace reference [3] in the second sentence here …in the central nervous system [e.g. PMID: 35682795 and 33515537].
Answer: Changed as suggested: 53
7) Introduction (line 48-49): Merge this very short paragraph with the next one to much better define structure of PAX6 and PAX6(5a) and differences in their DNA-binding mechanisms.
Answer: A sentence added with reference: 57-59
8) Introduction (line 58): Current ref [7] only talks about GST-PD(5a), studies of PD, HD, PD/HD and PD95a)/HD.
Answer: A reference added: 65
9) Reformate the final manuscript to prevent splitting Table 1 into two pages. It is also possible to eliminate the final column (Reference) and state ref [11] within the heading.
Answer: Table 1 was rearranged as suggested.
10)M&M (line 114): You may consider this link as well: https://clinicalgenome.org/affiliation/40072/
Answer: Thank you for the kind suggestion. A use of advanced criteria for evaluating pathogenicity is not our scope of our report. The reference is left.
11)M&M (line 133): State explicitly that the reporter contains a single copy of DNA sequence from ref [7]
Answer: The sentence was revised as suggested with the re-numbered reference: 154
12)Reformate the final manuscript to prevent splitting Figure 2 into two pages.
Answer: Revised: page 9
13)Subsection 3.2.2.: Explain in detail that a wide range of synthetic and natural reporters were used in similar studies but that there is no known direct target of PAX6 that is expressed in retinal cells forming the fovea. Since the 5aCON-luc is activated by both PAX6 and PAX6(5a), it can be used for the purpose of the present study.
Answer: Thank you for your comment. In this study, only PAX6(5a) was analyzed. However, differences in the target genes of PAX6 and PAX6(5a) may alter cell phenotypes, so we plan to incorporate your suggestions into our future investigations.
14)Subsection 3.2.4.: Explain the current method and compare with more traditional gel shift assay.
Answer: A paragraph was added: 385-398
15)Discussion (last paragraph): Other limitations include lack of knowledge of genes directly and indirectly dysregulated by PAX6 and PAX6(5a) in human fovea.
Answer:  Thank you for the important comment. We have integrated this comment in the limitation: 474-475
16)Provide list of Abbreviations.
Answer: The Instructions for Authors of this Journal does not indicate the suggestion.